# Tracking Quantitative Characteristics of Cutting Maneuvers with Wearable Movement Sensors during Competitive Women’s Ultimate Frisbee Games

**DOI:** 10.3390/s20226508

**Published:** 2020-11-14

**Authors:** Paul R. Slaughter, Peter G. Adamczyk

**Affiliations:** 1Department of Mechanical Engineering, University of Wisconsin, Madison, WI 53706, USA; peter.adamczyk@wisc.edu; 2Department of Biomedical Engineering, University of Wisconsin, Madison, WI 53706, USA

**Keywords:** inertial sensors, wearable technology, cutting, biomechanics, ACL injury, sports science

## Abstract

(1) Ultimate frisbee involves frequent cutting motions, which have a high risk of anterior cruciate ligament (ACL) injury, especially for female players. This study investigated the in-game cutting maneuvers performed by female ultimate frisbee athletes to understand the movements that could put them at risk of ACL injury. (2) Lower-body kinematics and movement around the field were reconstructed from wearable lower-body inertial sensors worn by 12 female players during 16 league-sanctioned ultimate frisbee games. (3) 422 cuts were identified from speed and direction change criteria. The mean cut had approach speed of 3.4 m/s, approach acceleration of 3.1 m/s^2^, cut angle of 94 degrees, and ground-contact knee flexion of 34 degrees. Shallow cuts from 30 to 90 degrees were most common. Speed and acceleration did not change based on cut angle. Players on more competitive teams had higher speed and acceleration and reduced knee flexion during cutting. (4) This study demonstrates that a lower-body set of wearable inertial sensors can successfully track an athlete’s motion during real games, producing detailed biomechanical metrics of behavior and performance. These in-game measurements can be used to specify controlled cutting movements in future laboratory studies. These studies should prioritize higher-level players since they may exhibit higher-risk cutting behavior.

## 1. Introduction

Ultimate frisbee (UF) is a popular sport played by three million people in the United States each year [1]. At the collegiate level, there are 499 schools with men’s teams and 327 with women’s teams that are registered to compete [2]. In a study of club sports in college, more people were hurt playing UF than nearly any other sport including soccer, basketball, baseball, and volleyball, and surpassed only by rugby [3]. Injuries from UF tend to occur on the lower limbs of athletes with 53% of frequent players experiencing knee injuries [4]. These injuries are often sustained by women who are injured twice as often as men in collegiate club UF [3]. Some of this discrepancy may be due to women having a higher risk of anterior cruciate ligament (ACL) injury in general: female athletes of other sports have 2–8 times higher rates of ACL injury than their male counterparts [5,6,7,8]. In a study of men’s semi-professional UF, ligament sprains in the knee were the fifth highest cause of injury, making up 3.3% of all injury cases observed [9]. It could therefore be predicted that for high level female players, these kinds of injuries could make up 7–20% of all injuries. Despite this high level of risk, UF has been little studied in clinical and biomechanical research [3,4,9,10].

The present study documented and analyzed players’ speed, acceleration, and knee joint angles during in-game cutting movements for female UF players. Female participants were recruited since they are at a greater risk of injury. Cutting was examined because of its high correlation to ACL tears. Cutting is the act of changing direction in a sharp, sudden motion. Cutting has been well studied in other sports [11,12,13], but rarely outside of laboratory conditions. Thanks to a new generation of motion capture technology, Inertial sensors can now record movement in the real world rather than in labs [14,15,16,17]. UF is the perfect sport for this kind of experiment because it involves frequent cutting to the point that its most common position is even called a “cutter”. It is also important to capture players in competition settings because in similar sports like basketball and soccer, three to nine times more ACL injuries take place during games than during practices [6].

To begin understanding cutting in UF, this study used wearable movement sensors to track and characterize in-game cutting movements. These movement characteristics will help examine the type and amount of risk to ACL injury involved in playing UF. The present analysis focuses on kinematic quantities measurable with current field-based wearables appropriate for in-game use—specifically, the speeds, accelerations and knee angles associated with each cut. Speeds and accelerations during cutting were chosen because of their link to higher ACL loading [18], which is considered a risk factor for ACL injury such as ACL rupture. Knee flexion was also examined to determine individuals’ levels of quadriceps dominance during cutting, another risk factor for ACL injury [19]. These movement characteristics could be used to design an appropriate in-lab study of more detailed biomechanics, such as knee valgus movement and multiaxial knee moment, or to interpret past data from in-lab cutting studies in the context of UF. Additionally, this study tested athletes on teams of three different levels and examined how team caliber impacts movement characteristics that may indicate relative ACL injury risk. Among different divisions of collegiate basketball and soccer, more advanced teams do not appear to have higher levels of ACL injury [20], but such data are lacking for UF.

It is the goal of this study to create a foundation for understanding the cutting movements of female UF players. Knowing the characteristics of players’ cuts will help to replicate real-world conditions in any future laboratory testing. This study also examined how UF players on different teams perform cuts to determine what kinds of leagues should be targeted for future study and intervention. The long-term vision is to drive the research that inform both collective and individual training and injury prevention practices to ensure UF remains a safe and enjoyable sport.

## 2. Materials and Methods

### 2.1. In-Game Data Recordings

Twelve (12) healthy female UF players gave their prior written informed consent to participate in this study. The study was performed in adherence to the rules of the Declaration of Helsinki and the protocol was approved by the University of Wisconsin-Madison Health Sciences Institutional Review Board (protocol 2018-0743, approved 30 July 2018). These participants played in 17 tournament games on three different teams (an advanced collegiate team, A; an intermediate collegiate team, B; and a competitive Club team) while wearing a lower body XSens MVN Awinda inertial measurement unit (IMU) system (seven segments: two feet, two shanks, two thighs, one pelvis; XSens, The Netherlands; Figure 1). These sensors had nine degrees of freedom, collecting 3-axis angular velocity (±2000 degrees/s), 3-axis acceleration (±160 m/s^2^), and 3-axis earth magnetic field (±1.9 Gauss). The wearable sensors transmitted wirelessly in real-time to a wireless base-station connected to a laptop computer, which were carried along the sideline by the experimenter to stay within range of the instrumented player (roughly 50 m). The sensors have a 10 s buffer that allows for brief out-of-range periods as well. Data for three games (games 3, 6, and 12 in Appendix B, Table A1) were collected at 60 Hz with all other games collected at 100 Hz due to a software update.

Participants ranged in age from 20–34 years (mean 26 years) and wore cleats during all data collections. All players were in trained competitive condition and were not suffering from any current injury that impeded their ability to compete. The IMU system was calibrated before each game, at half time, and any other time the body reconstruction became noticeably flawed (for example, if a sensor shifted). Games were played on different grass fields that were in good condition (not muddy).

### 2.2. Movement Reconstruction and Segmentation

Raw IMU data were processed through the human body model in the proprietary software (XSens MVN Analyze; Figure 1) to reconstruct foot contact with the ground, movement of the body around the playing field, pose of the instrumented lower body segments (pelvis, thighs, shanks, feet), and joint angles at each hip, knee and ankle, at each data frame. These kinematic data were then exported for post-processing in MATLAB (The Mathworks, Natick, MA, USA). Custom MATLAB scripts processed all exported files and created the statistical results of this study with some published scripts used to create Figures 4 and 5 [21], Figures 7, 9 and 10 [22].

Overall movement was broken down into individual footfalls of the left and right feet. Footfalls were identified using the exported foot contact data: a footfall was defined from initial contact to final contact of one foot with the ground. To further improve these contact data, gaps of contact lasting 150 ms or less between footfalls of the same foot were closed, as this is too short a time for a realistic swing phase. Then, contact periods lasting 50 ms or less were removed, as contacts that brief were assumed to be erroneous contact detections.

The primary source of whole-body cut kinematics was the pelvis segment’s position. Pelvis position data were fed through a 5-point median filter to remove spikes in the data. These spikes were caused by the proprietary IMU software adjusting for the uneven ground the participants played on and imperfect motion reconstruction. After smoothing, a coordinate system was defined for each footfall, using the initial and final horizontal position of the pelvis to define the X-axis and the world-frame vertical to define the *Z*-axis. Using the initial and final horizontal pelvis position, footfall speed $\left(s_{ff}\right)$ was calculated across each footfall, in each footfall’s local coordinate system.

### 2.3. Identification of In-Game Cuts

Potential cuts of interest were initially identified by looking for patterns in footfall speed ($s_{ff}$) data. For any sequence of footfalls to be called a cut, the sequence had to have at least two successive footfalls with a lower $s_{ff}$ than the previous footfall. The last of these slowing footfalls (the one with the lowest $s_{ff}$) was called the cut footfall, while any previous slowing footfalls were called approach footfalls. The cut footfall had to be followed by at least one exit footfall, where $s_{ff}$ increased with each footfall compared to the previous one. We allowed up to five approach footfalls if each continued to have a slower $s_{ff}$ than the previous footfall. These footfalls were denoted as C-5 through C-1. The cut footfall was denoted C-0. We allowed up to five exit footfalls if each subsequent $s_{ff}$ continued to increase, and these were denoted C + 1 through C + 5. An example cut showing this sequence of footfalls is shown in Figure 2.

If a series of footfalls met these $s_{ff}$ requirements, thresholds on minimum speed and direction change were applied (Figure 2), as slow movements and near-straight-line movements are not typically considered cuts. The approach speed ($s_{app}$) was defined as the horizontal speed of the pelvis in the first frame of the first approach footfall. The exit speed ($s_{exit}$) was defined as the horizontal speed of the pelvis in the last frame of the last exit footfall. The change in direction associated with a potential cut was defined as the cut angle (θ), calculated using the direction of the horizontal-plane velocity at the beginning of C-1 and the final direction at the end of C + 1 (a straight line has 0 degree cut angle). For a sequence of footfalls to be called a cut, both $s_{app}$ and $s_{exit}$ had to be greater than 2 m/s, the cut angle had to exceed 30 degrees in magnitude, and no footfalls within the sequence could exceed 2 s in duration. A sequence of footfalls was classified a cut if it met all these thresholds and the $s_{ff}$ pattern conditions.

### 2.4. Outcome Metrics

For each cut, a series of properties were used to characterize the cutting movement. All the properties used in defining the cut—number of approach steps, number of exit steps, approach and exit speeds, and cut angle—were included. The data frame with the minimum horizontal pelvis speed during the cut footfall was defined as the cut frame, with the value of the minimum speed denoted $s_{min}$. Then the average approach acceleration $\left(a_{app}\right)$ was calculated as the difference of the approach speed $s_{app}$ and the minimum speed $s_{min}$, divided by the elapsed time between these frames. Exit acceleration $\left(a_{exit}\right)$ was calculated using $s_{min}$ and $s_{exit}$ and the time between their respective frames.

For each footfall in a cut, behavioral and biomechanical outcomes were measured. Footfall speeds ($s_{ff}$) used to define cuts were included. Footfall acceleration ($a_{ff})$ was defined as the mean pelvis acceleration vector during each footfall, expressed in the footfall’s local coordinate system. Knee flexion (**α**) was recorded as the mean knee flexion angle during each footfall (defining a straight knee as 0 degree). Other common ACL strain indicators, such as knee moment, were not calculated because ground reaction forces could not definitively be calculated from the kinematic-only data collected. Example videos of the reconstructed kinematics during cutting maneuvers are included as Appendix A.

### 2.5. Statistical Analysis

To characterize the distribution of cutting movements, cut properties were gathered into histograms and summary statistics. The numbers of approach steps and exit steps in each cut were characterized by mean and standard deviation. Approach speed and exit speed were plotted as histograms in increments of 1 m/s. Cut angle was plotted as a histogram in increments of 30 degrees. The relationships between approach and exit speeds and cut angle were plotted as polar plots, with mean and standard deviation of speed for each 30 degrees bin of cut angle. The relationships between approach and exit accelerations and cut angle were plotted similarly. The relationship of acceleration to speed within cutting maneuvers was determined for approach and exit phases. Approach and exit acceleration were plotted against approach and exit speed, respectively, and best-fit linear relationships were determined using a robust fitting regression tool (MATLAB fitlm) to reduce the influence of outliers. Finally, the footfall accelerations during left and right footfalls were plotted for each footfall of a cut. For each footfall index (C-5 to C + 5), each footfall acceleration was plotted by foot (left or right), along with the mean acceleration for footfalls of each foot. The lateral accelerations attributable to left and right footfalls were compared using a two-tailed *t*-test.

To determine behavioral performance differences among the three teams (A, B and Club), data were segmented according to each player’s team and compared statistically using ANOVA and a Tukey’s Honest Significant Difference test. Cut speeds and cut accelerations were compared across each team. The mean and standard deviation of approach speed, exit speed, approach acceleration, and exit acceleration are graphed for each team, and for all cuts (combining all teams). For approach acceleration, the absolute value was taken to improve graph formatting (approach accelerations are negative). Mean footfall acceleration magnitude for each team was compared for approach footfalls (C-5 to C-1), cut footfalls (C-0), exit footfalls (C + 1 to C + 5), and all cutting footfalls (C-5 to C + 5). Mean knee flexion across different cut phases and teams was plotted similarly. All processing code, as well as the raw and MATLAB data files were uploaded onto IEEE DataPort.

## 3. Results

Data collection yielded 93 points of UF games. 14 points were removed from consideration due to faulty data, including all data from one participant whose sensor calibration failed. The remaining 79 points lasted a mean of 3.4 min, spanned 16 games, and included a mean of 273 steps per point. Both the B-team and Club team had four participants take part in this study, while the A-team had three. The B-team was the most tested with 39 points of gameplay being captured. The A team was sampled during 22 points and the Club team had 18 points recorded. These data yielded 424 detected cuts. Two cuts were removed because of unrealistic cut accelerations (> 20 m/s^2^) brought on by faulty data. These descriptive characteristics are summarized in Table 1 with individual characteristics summarized in Appendix B, Table A1.

Mean values of approach and exit speed, number of approach and exit steps, and cut angle are presented in Table 2. Mean approach and exit speeds are 3.4 and 3.7 m/s respectively. Cut angle, which spans from 30 to 180 degrees, has a mean of 94 degrees. The numbers of approach and exit footfalls, which can range from 1 to 5, have means of 2.8 and 3.1, respectively.

To further characterize the distribution of cutting movements, frequency histograms for approach and exit speeds and cut angle are shown in Figure 3. The majority of cuts occurred at approach and exit speeds from 2 to 4 m/s. Average approach speed was 3.4 m/s and average exit speed was 3.7 m/s, and sprinting cuts (over 5 m/s) were few. Cut angles between 30 and 60 degrees were most common, though reversal cuts (150–180 degrees) were also prevalent. The mean cut angle was 94 degrees.

These two variables, cut speed and cut angle, are graphed together in Figure 4. Neither approach nor exit speed were significantly impacted by the cut angle; however mean approach and exit speed values were higher for the lower cut angles. The highest mean approach speed in Figure 4 is 3.60 m/s for 30–60 degrees, while the lowest is 3.28 m/s at 150–180 degrees. Mean exit speed ranged from 3.52 m/s at 150–180 degrees to 3.84 m/s for 30–60 degrees.

Similarly, there were no statistical differences in mean cut acceleration at different cut angles. In Figure 5, mean approach acceleration varied from 2.73 m/s^2^ at 120–150 degrees to 3.28 m/s^2^ for 90–120 degrees. Similarly, mean exit acceleration ranged from 2.55 m/s^2^ at 90–120 degrees to 3.09 m/s^2^ for 30–60 degrees. Outliers present in cut acceleration and speed data were caused by errors in the IMU system’s footfall detection, which shortened the time over which some accelerations were computed.

Though cut speed and acceleration were unaffected by cut angle, they were linked to each other. Figure 6 shows that faster cuts tend to have higher acceleration for both the approach and exit phases. The best fit line placed on the approach data yielded a slope of−0.68 s^−1^ (*p* < 0.001), a 95% confidence interval of −0.858 to −0.545, and an R^2^ of 0.20. For the exit data, the best fit line slope was 0.56 s^−1^ (*p* < 0.001), a 95% confidence interval of 0.417 to 0.639, and a R^2^ of 0.31.

Speed and acceleration during approach and exit differed across the three teams, as shown in Figure 7. Players on the A-team had significantly higher approach speed, approach acceleration, and exit acceleration than either the B-team or Club team. Players on the B-team had higher exit acceleration than those on the Club team. There were no statistical differences in final exit speed across the three teams, though their mean values followed the same trend as the other metrics. The mean A-team cuts began at 3.66 m/s and ended at 3.88 m/s. In contrast, the mean Club team cuts began at 3.28 m/s and ended at 3.55 m/s. Similar relationships were present in acceleration: the mean approach and exit accelerations for the A-team were 3.95 m/s^2^ and 3.53 m/s^2^, respectively, whereas for the Club team, the mean accelerations were 2.37 m/s^2^ for approach and 2.13 m/s^2^ for exit. For each of these mean speed and acceleration metrics, the B-team cuts were between the A-team and the Club team. 

To further examine accelerations during cutting maneuvers, acceleration during each footfall was graphed for each cutting footfall. The mean acceleration for each approach footfall was pointed in the opposite direction of the player’s movement (-X). Similarly, mean acceleration for each exit footfall was pointed forward, in the direction of the player’s movement (+ X). The cut footfall (C-0) had near-zero acceleration on average. In each approach footfall (C-5 to C-1) and in C + 1, the lateral component of footfall acceleration differed significantly between right and left footfalls (*p* < 0.012). For right footfalls, mean acceleration of the body was to the left, and for left footfalls, mean acceleration was to the right. These footfall accelerations yielded many outliers, attributable mainly to inaccurate timing of foot contact detection by the IMU system. Plots of the acceleration with each footfall are shown for the middle five footfalls of each cut in Figure 8. Plots for all footfalls in all cuts are included in Appendix B, Figure A1.

Comparison of footfall acceleration magnitude across the three teams revealed differences in some phases of the cutting maneuver (Figure 9). The A-team had significantly higher acceleration magnitudes than either of the other two teams when comparing approach footfalls and all pooled footfalls. For exit footfalls, the Club team had significantly smaller acceleration magnitudes. There were no statistical differences in cut step acceleration among the three teams. For all cutting footfalls, the A-team had a mean footfall acceleration of 5.49 m/s^2^, the B-team had 4.92 m/s^2^, and the Club team had 3.86 m/s^2^.

Further differences across teams were observed in the knee flexion data across the 3 cut phases. Mean peak knee flexion for all footfalls was 31.75 degrees for the A-team, 34.98 degrees for the B-Team, and 34.96 degrees for the Club team (Figure 10). The A-team had statistically lower knee flexion than the B-team or Club team for approach, exit, and all footfalls. The A-team also had lower knee flexion than the B-team during the cut footfall.

## 4. Discussion

### 4.1. Discussion of Findings

This study introduces a way of measuring and detecting cutting maneuvers and estimating whole-body and lower-limb kinematics on-field during competitive sports, with ultimate frisbee (UF) as a specific case. Using these methods, cuts were detected, and their kinematics were analyzed. These cuts occurred often (1.5 times per minute of gameplay) which supports the notion that cutting is frequent in UF. The speed data from these cuts allows for context to be compared between on-field cutting and lab-based studies. For example, studies in which cutting was performed at 3 m/s (e.g., [23]) were testing cuts in the 37th percentile with respect to the in-game cuts observed in this sample of collegiate women, while other studies that used 4 m/s (e.g., [24]) were sampling in the 72nd percentile. Similarly, cuts have been studied in the lab across the whole spectrum of direction change, from 45 to 180 degrees [24,25]. In this study, players averaged 90-degree cuts, although there is no overwhelming preference by the players. Though it is important for cuts of all speeds and direction changes to be studied, this data set shows how frequent certain kinds of cuts are in UF competition. By having participants run at different speeds and follow different cutting paths, representative speeds and direction changes can be created in future lab-based studies on UF players.

Accelerations seen on the field can also be replicated in labs. This could be done by adjusting the amount of space a participant may have to perform the approach or exit of a cut. Figure 4 and Figure 5 show that acceleration and speed are constant across different changes in direction. This contradicts what has been observed in previous lab studies, wherein greater changes of direction yielded lower speeds [24]. This disagreement may be due to players cutting with different amounts of effort, which impacts acceleration and speed [12]. his random variability may overshadow any differences between acceleration and direction change (θ) that a controlled, maximal-effort lab study could discover. Conversely, the observation that in real UF games there is no difference in acceleration for different cutting angles suggests that certain lab conditions may not be relevant for understanding true in-game injury risk.

Figure 6 shows that cut speed and cut acceleration are correlated to each other. This relationship was anticipated because both high speeds and high accelerations are associated with intense moments in a game. Faster cuts in laboratories should therefore also have greater cut acceleration. On the other hand, the acceleration vs. speed plots also have a great deal of variability, which may indicate the variety of strategies a cutting player may use to elude a defender or, on defense, to keep up with an opposing player trying to get free. Such strategies may include, for example, high-acceleration cuts from relatively modest speeds (surprise cuts) or high-speed cuts with relatively low acceleration (e.g., if the player is already free from her defender but needs to adjust direction to reach an open space or catch the disc). Thus, the overall relationship between speed and acceleration holds, but it is heavily influenced by behavioral factors.

Figure 8 provides insight into the kinds of accelerations these players are experiencing when cutting during each footfall. Left and right feet were separated and shown to have the same acceleration magnitude. However, the right foot acceleration was more leftward and the left foot acceleration more rightward during approach footfalls. This was expected because each leg tends to push the body toward the opposite side during locomotion. However, this relationship was not true for most exit cut steps: the lateral difference in acceleration disappeared after step C + 1. The reasons for lateral acceleration asymmetry during approach but not during exit are not fully clear and could be studied in greater detail in the future. One possible explanation is that players may use a wider stance during deceleration to leave themselves the option of cutting either direction; then after the cut, they may simply sprint in a straight line to maximize their chances of getting free. Regardless of the cause, these graphs show footfall accelerations which could be used as a target for future lab-based data. Since higher accelerations are linked with higher ground reaction forces (GRF) and ACL loading [12,18], matching this cutting metric is important in order to study ACL injury risks for any sport.

Team comparisons were made in order to determine how different levels of competitive play may impact cutting behavior. This can cleanly be done by comparing the A-team and B-team participants in this study. They were both part of a single collegiate organization which divided up into two teams based mostly on skill level and experience. Comparing these teams with the Club team is more difficult because it is hard to establish the relative level of the Club team players. The Club team was comprised of adults, only some of whom were college age. Anecdotally, the Club team appeared to the researchers more similar to the B-team, and this similarity is reflected in the data.

Figure 7 shows that the A-team had higher cut speeds and higher approach acceleration. Though exit acceleration had a higher mean than the other teams, this difference was not significant (*p* = 0.09). The observed cutting differences probably reflect the higher-caliber games the A-team plays in. Acceleration for each footfall, shown in Figure 9, indicates that the A-team has higher footfall acceleration ($a_{ff})$ for approach steps and exit steps. This further supports the idea that the A-team cuts with greater acceleration, and thus is applying greater horizontal force against the ground with each cutting step. This difference was expected because players on higher-level teams are likely to be more athletic and play in more competitive games. The A-team, and advanced UF teams in general, may be at a greater risk of ACL injury because high speeds and accelerations cause larger ACL loading when cutting [18]. 

Additional risk for the A-team is suggested by their small knee flexion angles observed during cuts (Figure 10). They had statistically less knee flexion overall (i.e., for all cutting steps) and specifically for exit steps when compared to either other team. For all cutting steps, the difference between the A and B teams was 3 degrees. In comparison, the knee flexion difference between male and female soccer players was shown to be 7−10 degrees during cutting [26]. Given the large differences between sexes with regards to ACL injury rates, a 3-degree difference could still result in more ACL strain for the A-team.

In collegiate basketball and soccer, different rates of ACL injury have not been found among different levels of competition [20]. It is therefore contradictory that the A-team appeared to exhibit higher-risk movements during cutting in this study. However, even if these movements result in higher ground reaction forces and ACL strain, it is possible this would not result in more injuries. For example, if A-team players have stronger knee-stabilizing muscles, perhaps they can endure these higher-risk movements. If this is the case, more competitive players may exhibit higher-risk behavior while not actually having higher rates of ACL injury. A wide survey study would have to be conducted to know if these faster and harder cuts performed by higher-level teams translated to more ACL injuries. Until this is performed however, the results of this study suggest that future studies and interventions should focus on players on more competitive teams since they exhibit higher risk kinematics while cutting in UF games.

### 4.2. Limitations

With the design of this study, there are components of cutting in UF that are left unexamined. First, comparisons between player positions (“cutter” vs. “handler”) were not performed. This is because players may switch positions depending on who else from their team is playing during a point. Additionally, the nominal position of the player covers only offensive possessions, whereas many of these sampled cuts are likely on defense, where a player’s cuts are likely to match more closely the position of the opposing player they are guarding. Therefore, the most common position a participant played was recorded, but each individual’s data may include cuts made in both handler and cutter positions and on defense. Comparisons between players’ nominal positions could be calculated in a future study with a greater number of players, subject to the same caveats. For similar reasons, it was deemed unnecessary to equalize the ratio of players in each position across the three teams. Avoiding this enrollment restriction enabled the study to enroll any player and thereby to maximize participation, but the consequence was that team comparisons were also made without accounting for the positions of the participants. Another limitation of the team comparisons is that they implicitly assumed that all players on a team have similar cutting behavior. This assumption allowed cuts from all players on each team to be pooled; however, a consequence is that the sample includes more cuts from some players than from others, which could bias the data. An alternative approach would be to limit the number of cuts analyzed from each player, but this approach would have discarded too many valuable measurements from the present limited data set. Finally, the finding that the A-team had greater ACL injury risk factors in this study assumes that the only difference among teams was their level of competition. Other situational factors could have played a part in differentiating the teams, such as weather or field conditions, the importance of the captured game, or random error based on who participated from each team. These results, again, could be strengthened with more participants. Differences in age may also have led to differences between teams. The average age of the club players was 30, potentially a bit removed from their physical prime. The A-team (average age 25) was older than the B-team (average age 21) which likely reflects differences in experience and may also affect physical maturity and strength.

More types of data could have been reported if the games had been filmed. Filming games would have allowed for cuts to be separated by possession, i.e., when the participant was on offense vs. defense. This would allow for comparisons to be made between unexpected (defensive) and expected (offensive) cuts, as in past studies [27,28]. Additional comparisons could also have been studied if injury history data had been collected. Players had no injuries that were currently limiting their ability to play (this was an inclusion criterion), but it is possible that some may have had previous injuries that could still impact their cutting mechanics and have affected these results.

The most important shortcoming of measuring cutting during games is that the wearable inertial motion capture suit measures only kinematics, rather than the combined kinematics and kinetics (e.g., ground reaction forces (GRF), joint moments (JM)) measured in a motion laboratory. At present, there are no feasible approaches for complete biomechanical analysis based on wearables. Partial information can be obtained from in-shoe pressure insoles, but these do not measure the full GRF vector. Alternatively, wearing a full body IMU suit could allow GRF and JM to be estimated [29]. However, after a few trial data collections, feedback from participants revealed the upper-body sensors were obtrusive. This study chose to minimize interference with participants’ normal game play, so only IMU’s on the lower body were used. Another recent technique could allow direct estimation of joint moments through tendon forces [30], but a compact wearable version suitable for in-game recording is not yet available. Thus, current wearables are not yet sufficient to fully replace a laboratory.

Errors with the IMU system made 16 points of play unusable. These faulty data were either corrupted (files unable to be opened) or contained obviously faulty kinematic reconstructions of the player. These instances could be brought about by poor system calibration, sensors going out of range, or sensors falling off the participant during the game. The IMU software was also imperfect when detecting ground contact. This led to some accelerations registering during the supposed flight phase. Though contact with another player on the field may have produced some of these effects, it is unlikely player contact is the entire cause of the errors. These footfall inaccuracies caused most of the data point outliers that were either not shown in graphs or removed from the data set. Additionally, although valgus knee rotation is a risk factor for ACL injuries, knee valgus rotations from the IMU software were not reliably calculated, with unrealistic spikes in rotations populating the data set. It was therefore excluded from this paper.

### 4.3. Future Work

In addition to laboratory studies, there are many possibilities for future UF research to take place on the field. Landing maneuvers, as well as cutting, have been shown to cause ACL injuries [12,31,32] and could be studied using this existing data set. The cuts characterized in this study could also be further classified into the 3 types of cutting techniques (side-step, crossover, split-step) which have previously been compared in lab settings [13]. Examining how cuts change throughout the duration of a game or tournament may also be valuable since fatigue can increase risk of ACL injury [33]. Filming games would also be useful for examining planned (offense) vs. unexpected (defense) cutting maneuvers. Additionally, GRF estimation methods using a limited number of sensors [34] could be further developed. These forces and moments would allow other important injury risk metrics to be calculated while not burdening the participant with awkward sensors. If a GRF estimation method is validated using only a set of lower-body IMU sensors, this UF data set could be modeled in OpenSim or other biomechanical modeling software. Such a model could measure ACL strain for detected cuts and determine how this strain could be affected by strengthening different muscles [35]. Armed with these findings, physical therapists and athletic trainers could create exercise programs specifically designed to reduce ACL injuries during cutting for individuals or UF athletes as a whole.

## 5. Conclusions

Current wearable movement sensors provide accurate enough estimates of lower body kinematics to characterize important aspects of athletic maneuvers such as cutting, while being unobtrusive enough to wear during competition in some sports. The ability to measure not just accelerations and step counts, but also foot placement, direction changes and some aspects of leg kinematics during cutting maneuvers enables the prevalence and risk level of dynamic maneuvers to be characterized. Further methodological development should emphasize estimating ground reaction forces and lower-limb joint kinetics using a limited set of sensors, and robust motion reconstruction in the highly dynamic conditions that occur in sports.

Cutting is shown to be a common maneuver in women’s UF games, occurring roughly 1.5 times per minute during play. Cuts occur in all directions at a range of starting and ending speeds, most prominently the 2 to 4 m/s range, with cut angles of 30 to 90 degrees and reversal cuts (150 to 180 degrees) most common. These kinematic characteristics will serve to inform future studies that strive to replicate in-game cutting movements in laboratory conditions. Acceleration, speed, and knee flexion calculated for these game-time cuts serve to quantify movements associated with ACL injury risk for UF players. Players on higher-level teams appear to cut from faster speeds and with higher accelerations, and to exhibit lower stance-phase knee flexion during these maneuvers, which could lead to greater risk of ACL injuries for those athletes. Future studies of cutting mechanics and interventions should place priority on these more advanced teams, while also seeking to understand the prevalence of ACL injuries and risk factors and protective factors at different levels of UF play.

## Figures and Tables

**Figure 1 sensors-20-06508-f001:**
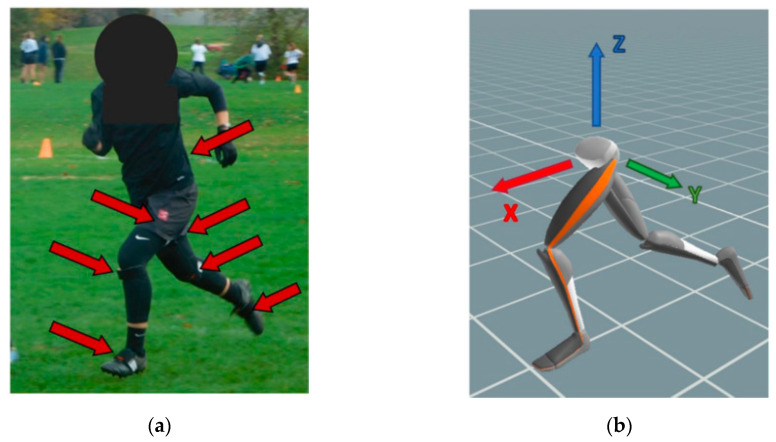
(**a**) Shows a participant with the seven IMU sensors. (**b**) is the kinematic model created from their movement and the coordinate system defined at the start of each footfall.

**Figure 2 sensors-20-06508-f002:**
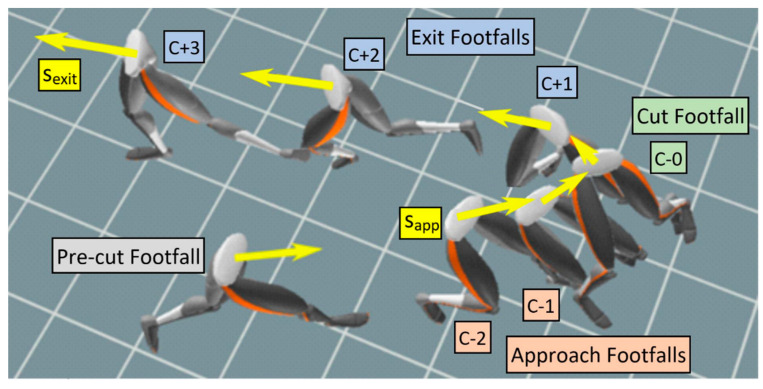
The 3 phases of cuts are the approach footfalls, the cut footfall, and the exit footfalls. The start of each footfall is shown. The yellow arrows represent the $s_{ff}$ of each cut step (not drawn to scale).

**Figure 3 sensors-20-06508-f003:**
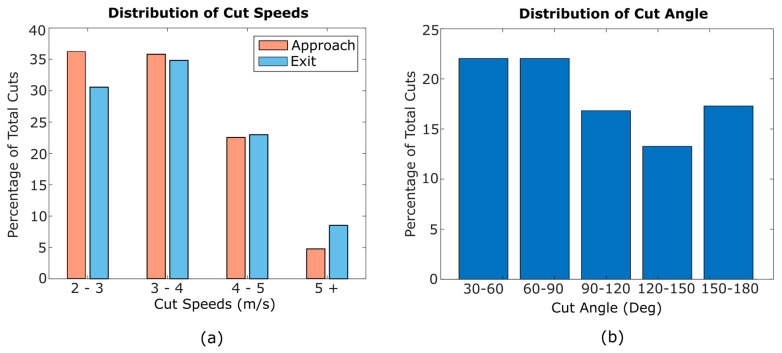
Histograms showing the distribution of cuts across speed (**a**) and cut angle (**b**).

**Figure 4 sensors-20-06508-f004:**
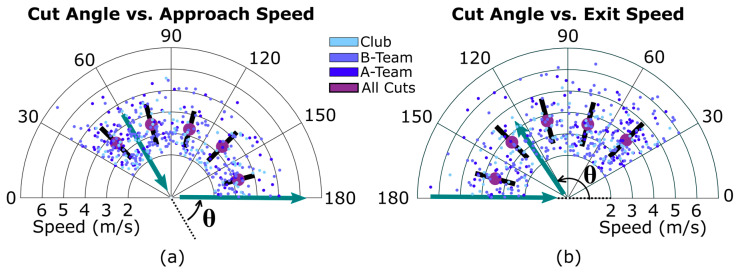
Distribution of approach (**a**) and exit (**b**) speeds for different cut angle ranges (in degrees). The large purple dot shows the average speed for each cut angle range, while the black line shows standard deviation. Speeds from the three teams are shown in different colors, and an example path of a player through a cut is shown in green arrows. The polar axes are restricted to better show trends; 4 outlier speeds are outside this view.

**Figure 5 sensors-20-06508-f005:**
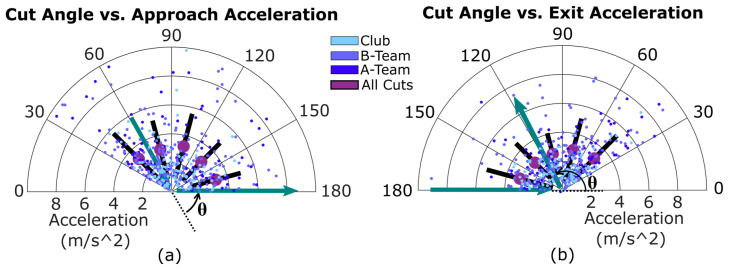
Approach (**a**) and exit (**b**) accelerations for different cut angle ranges (in degrees). The purple dot shows the average speed for each cut angle range, while the black line shows standard deviation. Though approach acceleration values are all negative, they are shown on the left as positive. Accelerations from the three teams are shown in different colors and an example path of a player through a cut is shown in green arrows. The polar axes are restricted to better show trends; 13 outlier accelerations are outside this view.

**Figure 6 sensors-20-06508-f006:**
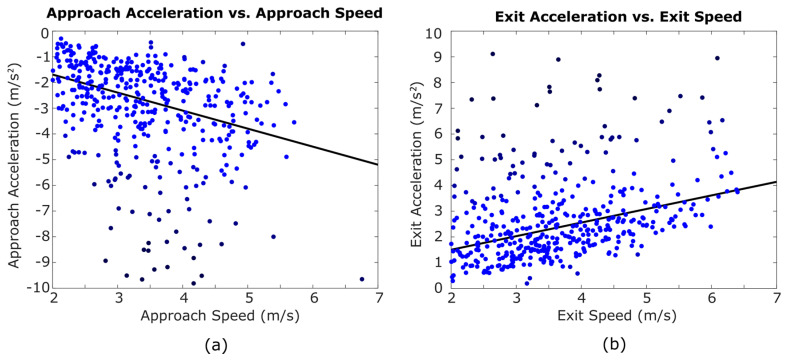
A linear regression model was applied to each data set to generate trend lines. Both approach (**a**) and exit (**b**) accelerations increased in magnitude with higher approach and exit speeds. Iterative weighted robust fitting was used to reduce the influence of outliers. Outliers were shaded darker than data closer to the trend line. 13 acceleration and 4 speed outliers are outside this view.

**Figure 7 sensors-20-06508-f007:**
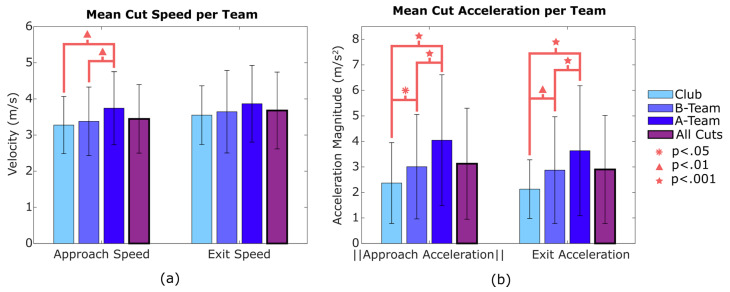
Mean and standard deviation for speeds (**a**) and accelerations (**b**) in approach and exit footfalls for each team. Pooled means for all cuts observed (across all teams) are also included. The A-team players performed cuts with greater speed and acceleration. Approach accelerations are shown as absolute values (raw values are negative).

**Figure 8 sensors-20-06508-f008:**
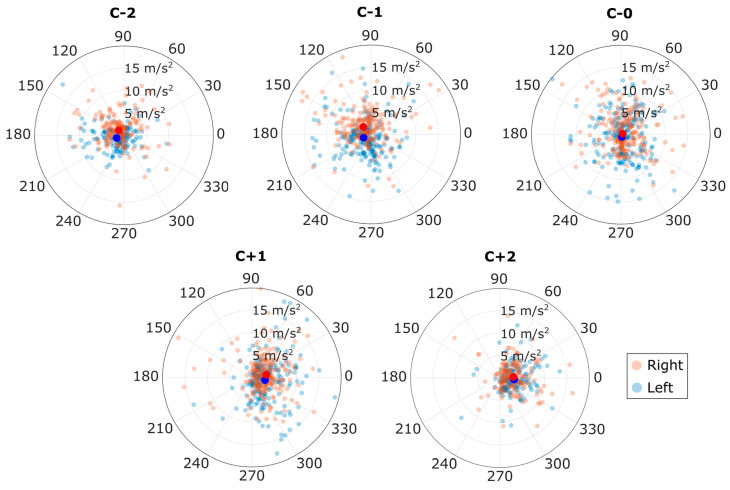
Footfall accelerations for each cutting footfall. The zero-degree direction was defined as the player’s local forward axis (+ X); the 90°line (+ Y) is to the player’s left. The solid blue and red dots represent the mean left and right accelerations, respectively. The polar axes are restricted to 20 m/s magnitude to better show trends; 24 outlier footfalls are outside this view.

**Figure 9 sensors-20-06508-f009:**
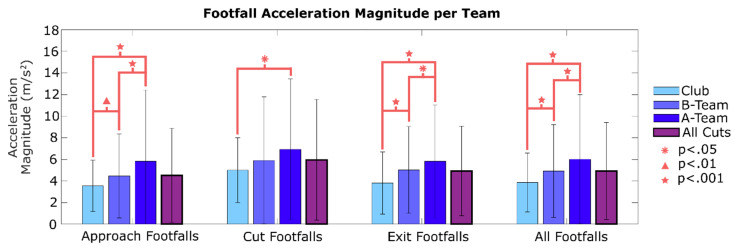
Mean footfall acceleration magnitude for each team, and for all cuts collectively. Mean values for each phase of the cut, in addition to all cutting footfalls, are shown. Cuts from the A-team tend to have larger footfall acceleration vectors.

**Figure 10 sensors-20-06508-f010:**
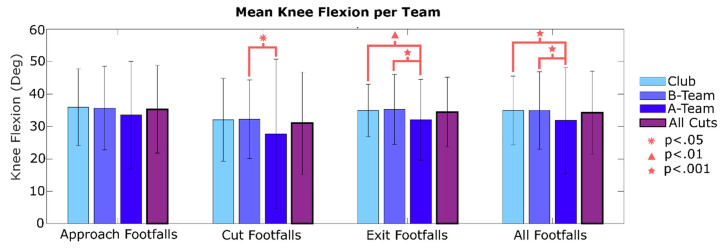
Mean knee flexion during ground contact for cutting footfalls for each team, and for all cuts collectively. Mean values for each phase of the cut, in addition to all cutting footfalls, are shown. A-team players tend to cut with lower knee flexion throughout the cutting maneuver.

**Table 1 sensors-20-06508-t001:** Total values for each team.

Team	Participants	Games	Points	Cuts
A	3 (3 cutters)	6	22	104
B	4 (2 cutters, 2 handlers)	6	39	227
Club	4 (3 cutters, 1 handler)	4	18	91
*Total*	*11*	*16*	*79*	*422*

**Table 2 sensors-20-06508-t002:** Mean values of basic cut metrics for all detected cuts.

Approach Speed	Approach Footfalls	Cut Angle	Exit Footfalls	Exit Speed
3.4 ± 0.9 m/s	2.8 ± 1.5	94 ± 45 degree	3.1 ± 1.6	3.7 ± 1.1 m/s

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
