# Peer review of "Tracking Quantitative Characteristics of Cutting Maneuvers with Wearable Movement Sensors during Competitive Women’s Ultimate Frisbee Games"

_sensors, 2020, doi:10.3390/s20226508_

Round 1
Reviewer 1 Report
In their research, the authors focused on elaborating a set of features describing cutting motions in the ultimate frisbee game. Special attention was paid to women, and within such a group, the studies were conducted.
The important aspect of these studies is their natural environment for the analyzed game type. The considered motions were extracted from the registered by sensors data, and some related metrics such as velocity, acceleration, and angle were calculated and compared. The differences in metric values were studied in terms of players' skills as well. Statistical analysis was performed, which did not confirm the statistical significance of some of the outcomes.
Although the aim of the studies was reached to some extent, the work has some weaknesses. They were pointed out in the paper, yet it seems worth emphasizing them, as they may impact the obtained results. At first, there is a lack of information regarding the conditions in which the experiment was performed. Did all groups have the same playing field? Were the atmospheric conditions the same? Additionally, the study also lacks reference to the players' physical conditions, which may impact motion characteristics.
Moreover, the authors suggest applying the results for organizing appropriated training and providing methods for injury prevention; however, it has not been indicated which of the analyzed features may generate injuries and the detection of which may indicate the proximity of an injury.
Reviewer 2 Report
The authors present a courageous step into a feasible approach to monitor , using wearable sensors, the body behavior of a competitor.
I have the following observations:
Obs.1. Although well known the ACL (anterior cruciate ligament) abbreviation must be explained.
Obs.2. The authors must somehow explain why there are differences between the composition of the analyzed teams A and B, in terms of cutters and handlers.
Obs.3. A brief presentation of the used sensors (for example type and performances) and the acquisition system is recommended.
Obs.4. The research can be completed with some input related to: game experience, the existence or not of previous injuries, the average age of the teams, the average height of the athletes in each team or even average weight. I expected the teams to play on the same league, to compare them in a complete way, although one of them seems to be more physically suited.
Reviewer 3 Report
When analyzing the article, you can observe:
- No description of the measurement methodology. What did the authors measure: acceleration?
- The authors did not write what and how they measured. What sensors did they use? How did they record the data?
- The authors have no experience in writing this type of article.
- There is no scientific contribution in the article.
Round 2
Reviewer 1 Report
Dear Authors, thank you for addressing my comments.
Reviewer 3 Report
The suggested modifications have been satisfactorily made.